# A light carbon isotope composition for the Sun

James R. Lyons[1], Ehsan Gharib-Nezhad[2] & Thomas R. Ayres[3]

Measurements by the Genesis mission have shown that solar wind oxygen is depleted in the rare isotopes, $^{17}O$ and $^{18}O$, by approximately 80 and 100‰, respectively, relative to Earth's oceans, with inferred photospheric values of about −60‰ for both isotopes. Direct astronomical measurements of CO absorption lines in the solar photosphere have previously yielded a wide range of O isotope ratios. Here, we reanalyze the line strengths for high-temperature rovibrational transitions in photospheric CO from ATMOS FTS data, and obtain an $^{18}O$ depletion of $\delta^{18}O = -50 \pm 11$‰ ($1\sigma$). From the same analysis we find a carbon isotope ratio of $\delta^{13}C = -48 \pm 7$‰ ($1\sigma$) for the photosphere. This implies that the primary reservoirs of carbon on the terrestrial planets are enriched in $^{13}C$ relative to the bulk material from which the solar system formed, possibly as a result of CO self-shielding or inheritance from the parent cloud.

[1] School of Earth and Space Exploration, Arizona State University, 781 S. Terrace Rd, Tempe, AZ 85281, USA. [2] School of Molecular Sciences, Arizona State University, Tempe, AZ 85287, USA. [3] Center for Astrophysics and Space Astronomy, University of Colorado, Boulder, CO 80309, USA. Correspondence and requests for materials should be addressed to J.R.L. (email: jimlyons@asu.edu) or to E.G-N. (email: e.gharibnezhad@asu.edu) or to T.R.A. (email: Thomas.ayres@colorado.edu)

The light stable isotope compositions of meteorites, planets, and the Sun constrain how our solar system formed and the nature of the formation environment. Given that the protosun was the primary mass of the nascent solar system, its isotopic composition is of particular importance. The NASA Genesis mission succeeded in measuring O and N isotope ratios in returned solar wind samples[1,2]. Extrapolation of the solar wind results to the solar photosphere, by accounting for isotope fractionation due to ion collisions in the corona[3], has demonstrated that Earth's silicates, and by extension all terrestrial planet silicates, experienced a very different isotopic history compared to the bulk Sun. This result was predicted based on O isotope data from inclusions in primitive meteorites[4,5], and it was suggested that photochemical processes in the solar nebula were responsible for the isotopic difference between the bulk Sun and planetary materials[5–7].

Carbon isotopes are also of central importance to understanding solar system formation, and are essential to interpreting kinetic isotope fractionation in biochemical systems. Genesis has not yet reported C isotope values for the solar wind, due in part to the composition of the concentrator targets (SiC and diamond-like C). Measurement of solar wind implanted in lunar regolith silicate grains yielded a $\delta^{13}C \sim -105 \pm 20‰$, from which a bulk solar ratio of $\sim-150$ to $-100‰$ was inferred[8] (C isotope $\delta$-values are computed relative to the Vienna Pee Dee Belemnite (VPDB) standard with $^{13}C/^{12}C = 0.0112372$. All errors are $1\sigma$, unless stated otherwise). Ion microprobe measurements[9] of TiC grains in a CAI in the Isheyevo meteorite found $\delta^{13}C = 1.1 \pm 7‰$, a value indistinguishable from Earth mantle C. TiC and CAIs are high-temperature condensates, so it was argued that the TiC isotope ratio represents the bulk solar nebula, and therefore the bulk Sun. Measurement of N isotopes in TiN grains in the same Isheyevo CAI yielded $\delta^{15}N = -359 \pm 5‰$ relative to atmospheric $N_2$ ($^{15}N/^{14}N = 3.676 \times 10^{-3}$), a value consistent with the bulk solar value inferred by Genesis[2]. Observations of CO absorption in the solar photosphere yielded C isotope ratios either consistent with terrestrial, $\delta^{13}C \sim -30 \pm 45‰$ (ref. [10]), or highly enriched relative to terrestrial, $\delta^{13}C = 110 \pm 14‰$ (ref. [11]). Thus, previously reported C isotope ratios for the Sun and bulk solar nebula span a range of $\sim200‰$.

In order to resolve the discrepancies between astronomical observations of the photosphere and ion microprobe measurements of C isotopes in solar wind and TiC, we will first focus on solar O isotopes. Earlier measurements of the oxygen isotope ratios of the solar photosphere yielded ratios approximately consistent with terrestrial value ($\delta^{18}O \sim 41 \pm 59‰$; ref. [10]), or highly enriched in the rare isotopes ($\delta^{18}O \sim 130 \pm 15‰$; ref. [11]). Given the complexity of the solar atmosphere, and the potential for possible additional fractionation processes, it is important to reconcile the astronomically determined O isotope ratios of the photosphere with those inferred from Genesis. A recent reanalysis of shuttle-based observations of CO in the photosphere made significant progress to that end[12] by eliminating line blends of CO isotopic species with the tails of $^{12}C^{16}O$ lines, and by using a 3D hydrodynamic model atmosphere[13] of the photosphere to properly account for temperature variations within the footprint of the observations. However, the literature values for the oscillator strength ($f$-value) scale for CO rovibrational transitions left a 60‰ range of uncertainty, spanning the $^{18}O/^{16}O$ ratios from terrestrial values to those inferred for the photosphere from Genesis, precisely the range of most interest[12]. We resolve these differences in $f$-values, and present new photospheric O isotope ratios. We then derive a self-consistent $^{13}C/^{12}C$ ratio for the photosphere, which defines the C isotope ratio for the initial solar system.

## Results

### ATMOS Fourier transform spectrometer data and solar atmosphere model.

CO rovibrational transitions dominate the 2–5 micron spectral region of the photosphere. CO absorption line data were collected by the shuttle-borne ATMOS Fourier transform spectrometer (FTS) in the mid-1990s (ref. [14]). The ATMOS FTS data contain thousands of CO fundamental ($\Delta v = 1$) and first-overtone ($\Delta v = 2$) lines ($v$ is the vibrational quantum number) recorded at high signal-to-noise ratio ($\sim10^2$–$10^3$) and at high spectral resolution ($\omega/\Delta\omega \sim 150,000$)[14]. Because of the high temperature of the photosphere and chromosphere, most of the transitions are between highly excited vibrational states (i.e., 'hot' bands). We define the lower energy level, also called the excitation energy, as $E_{low}$. To convert the ATMOS data into isotopic abundances in the photosphere, both highly accurate line strengths for CO isotopologues, and a physically representative photospheric model are needed. We have closely followed the radiative transfer and solar atmosphere modeling of Ayres et al.[12], who constructed hybrid line profiles by co-adding absorption lines of CO isotopologues with similar excitation energy ($E_{low}$), wavenumber range, and absorption depth (Fig. 1). Overlap of line tails from the main isotopologue $^{12}C^{16}O$ onto the line peaks of the rare isotopologues was avoided, removing a major source of systematic error for the isotopologue abundances. Additional details of the ATMOS FTS data and the method for analyzing the CO absorption lines are given in Methods.

The CO5BOLD 3D radiation hydrodynamic model of the solar photosphere is used to capture convection-related temperature variations (both horizontal and vertical) associated with solar granulation[13]. Sixteen snapshots from the 3D hydrodynamic atmosphere model are used to capture the temperature variation associated with convection at the base of the photosphere, and to quantify the uncertainty in derived isotope abundances associated with the hydrodynamic model, as described in Ayres et al.[12].

### CO rovibrational spectroscopy.

For a given rovibrational transition, involving a lower level ($v''$, $J''$) and an upper level ($v'$, $J'$),

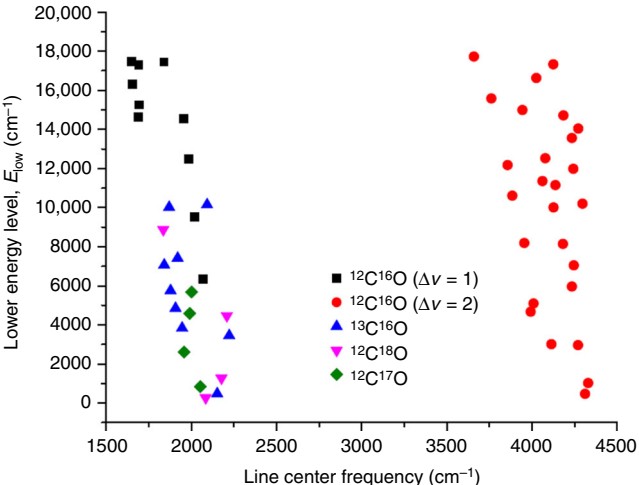

**Fig. 1** Lower energy levels versus line center frequency for co-added lines. Lines are constructed from ATMOS FTS data for all CO isotopologues analyzed[12]. The lower energy level is the energy of the lower state for a given transition. Most co-added lines consist of 3–6 individual lines of similar lower energy level and line center frequency. Overtone lines ($\Delta v = 2$, where $v$ is the vibrational level) are used only for $^{12}C^{16}O$. Analyses of rare isotopologues use only the fundamental transitions ($\Delta v = 1$). Comparison of the $^{12}C^{16}O$ abundance determined from $\Delta v = 1$ and $\Delta v = 2$ lines is used to make small corrections to the photospheric temperature profile

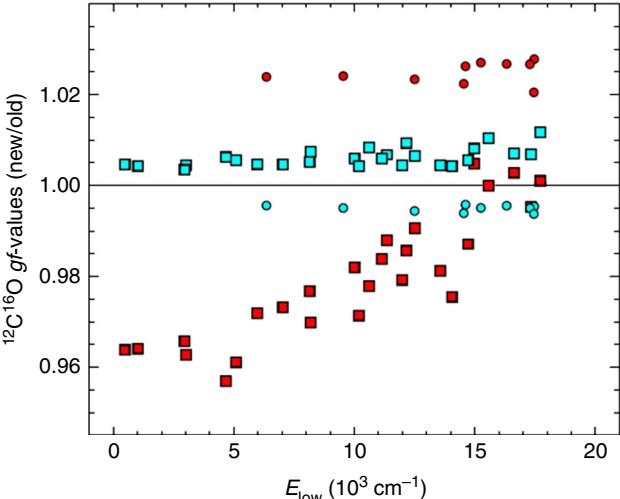

**Fig. 2** New $^{12}C^{16}O$ oscillator strengths. Strengths use the Li et al. dipole moment function[16] for the co-added lines shown in Fig. 1. The oscillator strengths are shown multiplied by the multiplicity, $g = 2J'' + 1$ for lower state rotational quantum number $J''$, and are therefore designated as *gf*-values. The new *gf*-values are shown normalized to previously published ("old") *gf*-values, where blue symbols are normalized by HR96[15], and red symbols are normalized by G94[18]. Circles are for $\Delta v = 1$ transitions, and squares are for $\Delta v = 2$. This figure shows that the new *gf*-values are considerably closer to the *gf*-values of HR96[15]. The ratio of *f*-values moves in the opposite direction from $\Delta v = 1$ to $\Delta v = 2$ for HR96 and G94 *f*-values, causing the 5–6% difference in photospheric isotope ratios previously found[12]

where $v$ and $J$ are the vibrational and rotational levels, the oscillator strength (or *f*-value) for the transition is given by[15]

$$f^{v'J'}_{v''J''} = \frac{8\pi^2 m_e}{3he^2} \frac{\sigma}{2J'' + 1} S_{HL} \left| M^{v'J'}_{v''J''} \right|^2. \qquad (1)$$

Here $\sigma$ is the frequency of the transition (line center frequency of Fig. 1), $M$ is the rovibrational dipole moment, and $S_{HL}$ is the Hönl–London factor, which has a value of $J''$ for P branches and $J'' + 1$ for R branches. We employ a new dipole moment function[16] and a spectroscopically determined potential energy function[17] for the electronic ground state of CO to calculate a new set of vibration-rotational *f*-values for $^{12}C^{16}O$, $^{12}C^{17}O$, $^{12}C^{18}O$, and $^{13}C^{16}O$ isotopologues.

Figure 2 shows the $^{12}C^{16}O$ *f*-value ratios of the new values compared to the old values of Hure et al.[15] (HR96) and Goorvitch[18] (G94). The 2–3% percent differences in the *f*-value ratios between the new *f*-values computed from Li et al.[16], and the HR96 and G94 *f*-values are of opposite sign for the fundamental and first overtone transitions, which accounts for about 50‰ of the 60‰ difference in the $^{18}O/^{16}O$ ratios obtained in Ayres et al.[12] for these HR96 and G94 *f*-value scales. This reduction in the uncertainty in *f*-values greatly reduces the uncertainty in our previously determined photospheric isotope ratios. The difference between the HR96 and G94 *f*-values is due to the dipole moment functions used. HR96 used a DMF from Langhoff and Bauschlicher[19] and G94 used Chakerian et al.[20], with the former showing better agreement to Li et al.[16]. The *f*-value ratios of Fig. 2 show only a slight dependence on isotope, and therefore the ratios for $^{12}C^{17}O$, $^{12}C^{18}O$, and $^{13}C^{16}O$ are quite similar to those of $^{12}C^{16}O$, and are not shown here[12,21]. The work of Li et al.[16] resolves a long-standing uncertainty in CO rovibrational *f*-values.

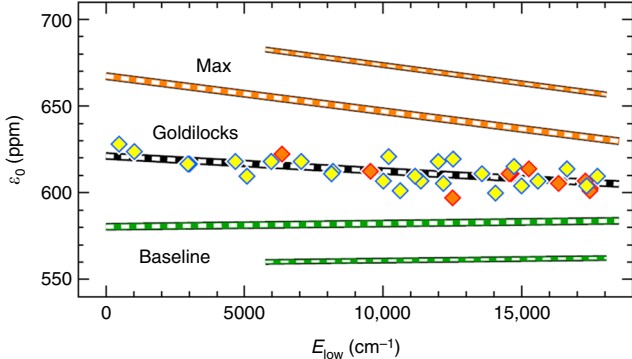

**Fig. 3** Elemental oxygen fraction from ATMOS $^{12}C^{16}O$ data and for three different photospheric models. $E_{low}$ is the energy of the lower rovibrational state for the transition. The "Goldilocks" model contains a mid-latitude photospheric temperature perturbation of 43 K that yields equality between the O fraction for $\Delta v = 2$ transitions (yellow diamonds) and $\Delta v = 1$ transitions (orange diamonds), with a linear fit to the data (black line). The Max (90 K) and Baseline (0 K) models yield less consistent results (small orange and green lines are for $\Delta v = 1$). (Figure after Ayres et al.[12])

**Photospheric temperature perturbation**. The solar hydro-dynamic model does not perfectly capture the temperature profile of the Sun. As done previously[12], three solar atmospheric temperature profiles are considered. One of the profiles, "baseline", is the mean temperature profile from the hydrodynamic model[13,22]. A second profile is a 90 K enhancement of the photospheric temperature, applied from ~$10^1$ to $10^4$ dyne cm$^{-2}$ (ref. [12]); i.e., from the middle photosphere to the lower chromosphere. The third profile, "Goldilocks", is a 43 K enhancement of the photospheric temperature. Figure 3 shows the dependence of the determined O fraction on the three temperature models. Only the Goldilocks profile yields consistent O abundance results for both fundamental and first overtone $^{12}C^{16}O$ transitions. It is important to emphasize that we have not "tuned" the O isotope results to agree with the Genesis inferred ratios. We have enforced agreement between O abundances derived from fundamental and overtone transitions by applying a temperature perturbation of 43 K; the resulting O isotope ratios are consistent with Genesis[1]. The slightly higher photospheric temperature may result from heating associated with magnetic fields or due to wave motion, neither of which are included in the radiative hydrodynamic model. Ideally, the O fraction would be identical for all lines, but a small trend is visible in Fig. 3, which may indicate the presence of additional small errors in the *f*-values or in model temperature profiles.

**Solar photosphere isotope ratios**. The C and O abundances are first determined from $^{12}C^{16}O$ lines, with the O abundance equal to twice the C abundance, $\varepsilon_O = 2\varepsilon_C$ (ref. [23]). Isotopic abundances are then computed separately for a given $^{16}O$ abundance. The derived O abundance is 605–620 ppm, which is slightly low compared to the preferred value of 640–680 from helioseismology[24]. The derived isotope ratios are weakly dependent on the absolute O abundance, and we do not expect that this small elemental disparity will significantly affect our O isotope ratios. Our $^{18}O$ abundance for the temperature-enhanced photosphere is $\delta^{18}O_{SMOW} = -50 \pm 11$‰, which is the same within errors as the inferred ratio from Genesis (Fig. 4). Our $^{17}O$ value is $\delta^{17}O_{SMOW} = -65 \pm 33$‰, which does not distinguish between the Genesis photosphere value and a terrestrial value at the $2\sigma$ level due to the low signal to noise ratio (SNR) of the $^{12}C^{17}O$ lines. Our results provide the first accurate, directly determined $^{18}O/^{16}O$ isotope ratio for the solar photosphere, and provide support for the

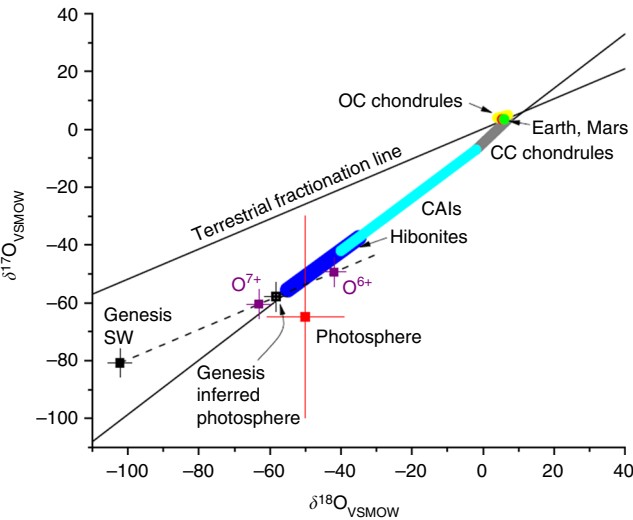

**Fig. 4** Directly measured oxygen isotope ratios of the photosphere from ATMOS observations of CO. Genesis solar wind and inferred photosphere values also shown, together with theoretical mass-dependent fractionated values for inefficient Coulomb drag for $O^{6+}$ and $O^{7+}$ ions in the solar corona[42] (purple squares). The value for $\delta^{18}O$ of the photosphere determined here (red square) is consistent with the inferred value from Genesis (black squares), and is distinct from a terrestrial value. All error bars are $1\sigma$. OC is ordinary chondrite (yellow); CC is carbonaceous chondrite (gray); CAIs are calcium-aluminum inclusions (light blue); hibonites (blue); bulk silicate Earth (green circle); bulk silicate Mars (red circle). The terrestrial fractionation line describes the array of O isotope values for the vast majority of materials on Earth; it has a mass-dependent slope of about 0.52

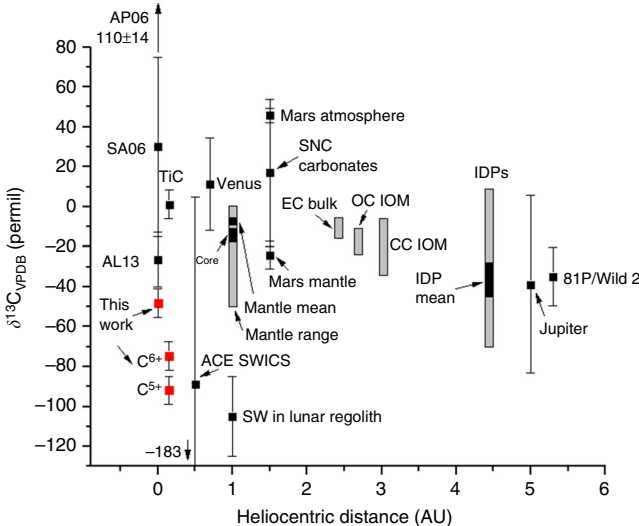

**Fig. 5** Carbon isotope ratios in the solar system. The value for CO in the Sun (red squares, $1\sigma$ errors) reported here is among the lightest in the solar system for bulk materials, and lighter than previously reported values, AL13 (ref. [12]) and SA06 (ref. [10]). The computed values for $C^{5+}$ and $C^{6+}$ ions in the solar corona[42] (in red) are similar to solar wind in lunar regolith[8] and to ACE SWICS measurements[29], but differ from the ratio in TiC from Isheyevo[9]. Other reservoirs include Earth mantle diamonds (mean and range)[26] and core estimate[58] (Earth atmosphere is about 2‰ lighter than mantle mean), Venus atmosphere[59], Mars atmosphere[60], Mars mantle from SNC silicates[61], and SNC carbonates[62]. Enstatite chondritic (EC) bulk C, ordinary chondrite (OC) insoluble organic matter (IOM), and carbonaceous chondrite (CC) IOM overlap with Earth mantle values[38]. The Jupiter and interplanetary dust particle (IDP) ratios spans a wide range[27,63]. Dust form comet 81 P/Wild 2 had ratios clustering in a fairly tight range[64]. The heliocentric distances shown for TiC, ECs, OCs, CCs, and IDPs are illustrative only. Additional reservoirs not shown here include Earth organic carbon, terrestrial and marine carbonate, minor organic components in carbonaceous chondrites, and various poorly constrained reservoirs such as Saturn, Neptune, and most comets[32]. It should also be noted that the fraction of C in Earth's core could be as much as 90% (ref. [65]), in which case the bulk Earth C isotope ratio would be approximately the core value

significant mass-dependent fractionation of O isotopes from the corona to the solar wind, most probably due to inefficient Coulomb drag (ICD)[3]. The predominant interpretation for the 60‰ difference in oxygen isotopes between the solar photosphere and the terrestrial planets is photochemical self-shielding of CO either in the solar nebula[5,7] or in the parent molecular cloud in which the solar system formed[6]. The near unity slope of the $\delta^{17}O$-$\delta^{18}O$ line defined by CAIs (Fig. 4) and of CO photolysis[5–7,25] is diagnostic of an abundance-dependent, rather than mass-dependent, fractionation process such as self-shielding.

Our analysis of the ATMOS CO data also yields the carbon isotope composition of the photosphere. For the same temperature profile, the photosphere is depleted in $^{13}C$ relative to terrestrial carbonates (VPDB standard) by $-48 \pm 7$‰ (Fig. 5). Mantle carbon is believed to have a mean $\delta^{13}C$ ~$-5$‰ (ref. [26]), suggesting that bulk terrestrial C is enriched in $^{13}C$ by nearly as much as bulk terrestrial O is enriched in $^{18}O$. Terrestrial planet atmospheres are all enriched in $^{13}C$ relative to the photosphere. Titanium carbide condensates in CAIs in the Isheyevo meteorite have $\delta^{13}C$ ~1‰ (ref. [9]), but the formation environment of these CAIs is unclear. Our photosphere ratio is near the $-40$‰ mean determined for Jupiter[27], but the large error bars encompass the terrestrial mean mantle ratio also (Fig. 5).

The $1\sigma$ error bars in Figs 4 and 5 include contributions from several sources, as summarized in Table 1 for the determined elemental abundance of oxygen, $\varepsilon_O$, and the three elemental isotope ratios reported here. Error 1 is the uncertainty due to scaling the 1st overtone O abundance ($\varepsilon_O$) to the O abundance derived from the less numerous fundamental transitions in the calculation of isotopologue abundances[12]. Error 2 is the internal sample uncertainty (standard error of the mean), which is the

dominant source of error for $^{16}O/^{17}O$ of the Sun, and which renders the $\delta^{17}O$ value to be of limited use in this work. Error 3 is the uncertainty due to the 'snapshot variability' in 3D hydro-dynamic model of the solar photosphere, as previously described[12]. Error 4 is the uncertainty in the oscillator strengths, taken to be ½ the difference between the results for HR96[15] and Li et al.[16]. Errors 1 though 4 added in quadrature yield the column "$1\sigma$ final" in Table 1, which for the isotope ratios correspond to the $\delta$-value errors reported above. If we include in the quadrature sum the full range of photospheric temperature perturbations (i.e., "Baseline" and "Max" from Fig. 3, with temperature enhancements of 0 and 90 K, respectively), we obtain Error 5 in Table 1. Error 5, which characterizes a solar atmosphere modelization error, dominates the other 4 error terms in all cases except $^{16}O/^{17}O$. However, this modelization error is removed by requiring that the $\Delta v = 1$ and 2 abundances for $^{12}C^{16}O$ are equivalent. We therefore reject the "Max" and "Baseline" results, and do not consider Error 5 in the reported error for O and C isotopes.

Gravitational settling of heavier isotopes into the solar radiative zone can slightly alter photospheric ratios over the age of the Sun. The convection zone has a vertically homogenous composition because of rapid turbulent mixing, while the

**Table 1 Mean and 1-sigma errors contributions for O abundance and isotope ratios in the Sun**

| Ratio | Mean | Error 1 | Error 2 | Error 3 | Error 4 | 1σ final | Error 5 | 1σ |
|---|---|---|---|---|---|---|---|---|
| $\varepsilon_O$ (ppm) | 612 | 0.0 | 1.4 | 1.6 | 6 | 6 | 30 | 31 |
| $^{12}C/^{13}C$ | 93.5 | 0.6 | 0.3 | 0.2 | 0.2 | 0.7 | 3 | 3.1 |
| $^{16}O/^{18}O$ | 525 | 3 | 5 | 1.4 | 1.1 | 6 | 20 | 21 |
| $^{16}O/^{17}O$ | 2815 | 17 | 103 | 7 | 6 | 105 | 100 | 145 |

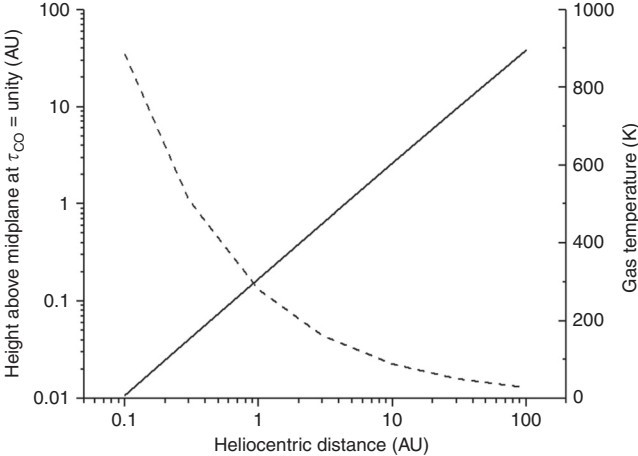

**Fig. 6** Height above midplane at which $\tau_{CO}$ = unity. Calculation assumes a CO absorption cross sections $\sigma_{CO} = 1 \times 10^{-16}$ cm$^2$ and a CO fraction (relative to H$_2$) $f_{CO} = 2 \times 10^{-4}$ (i.e., a H$_2$ column density of $10^{20}$ cm$^{-2}$). The corresponding H$_2$ number densities vary from ~$10^{10}$ cm$^{-3}$ at 0.1 AU to ~$10^6$ cm$^{-3}$ at 100 AU. Vertically isothermal gas temperature is also shown (dashed line). Calculations use an analytical disk model[34]

chemical composition and the isotopic ratios in the photosphere are expected to be very close to the convective zone for heavier elements like oxygen and carbon. Downward convective overshoot into the radiative zone reduces gravitational settling effects. If convective overshoot into the radiative zone is neglected, the depletions in convective zone and photospheric $\delta^{13}C$, $\delta^{15}N$, and $\delta^{18}O$ are −6.1, −4.4, and −9.1‰, respectively[28]. With convective overshoot present, the depletions in convective zone $\delta^{13}C$, $\delta^{15}N$, and $\delta^{18}O$ are predicted to be −4.3, −3.1, and −6.4‰, respectively[28]. These predicted isotopic shifts are non-negligible, and are mass-dependent, but fall within the 1σ uncertainties of the photospheric O and C isotope results presented here. For this reason we have not shown isotopic shifts due to gravitational settling explicitly in either Figs 4 or 5.

For comparison to solar wind measurements, our O and C isotope ratios must account for isotope fractionation during transport from the photosphere to the corona, an environment in which collisional Coulomb drag is believed to operate[3]. Computing fractionation due to ICD yields $\delta^{13}C$ values ~−75‰ and −95‰ for C$^{6+}$ and C$^{5+}$ ions, respectively (Fig. 5; "Methods"), which defines our predicted range for C isotopes in the solar wind. Reported values of C isotopes in solar wind implanted in lunar regolith grains[8] are ~ −90 to −120‰, which overlaps with our predicted solar wind value for C$^{5+}$ ions. The Advanced Composition Explorer Solar Wind Ion Composition Spectrometer (ACE SWICS) instrument[29] has also determined an isotopically light mean $\delta^{13}C$ value of ~−90‰, but with very large 1σ uncertainties (Fig. 5).

**Enrichment in terrestrial planet $^{13}C$.** The origin of the enrichment of the terrestrial planets in $^{13}C$ is a central question which, as for the enrichment of O isotopes in planetary materials, bears

on the formation environment of the solar system. We present here a preliminary evaluation of mechanisms that could produce enrichment of $^{13}C$ in the inner solar system. The $^{13}C$ enrichment could be a result of chemical processing in the solar nebula or parent molecular cloud, or could result from fractionation that accompanied accretion, differentiation, and atmospheric degassing during planet formation; here we focus on a nebular or parent cloud origin. Nebular C isotopes can be affected by many processes including CO self-shielding (as proposed for O isotopes), ion-molecule chemistry, CO ice formation in the outer nebula, and loss of CO in surface disk winds. Parent cloud C isotopes can be altered by similar processes, with the likely exclusion of surface winds.

Self-shielding during CO photodissociation, either in the solar nebula or parent molecular cloud, has been proposed[5–7] to be responsible for the enrichment of planetary materials in $^{17}O$ and $^{18}O$. Self-shielding was first observed in molecular clouds[30], and more recently has been measured in the oxygen isotopologues of CO in young protoplanetary disks[31]. In self-shielding, $^{12}C^{16}O$ lines are saturated at dissociating wavelengths (~100 nm), while $^{12}C^{17}O$ and $^{12}C^{18}O$ remain unsaturated due to their much lower column abundances, resulting in a massive enrichment of $^{17}O$ and $^{18}O$ from CO dissociation. Oxygen enriched in the heavy isotopes can be sequestered in nebular water ice in cool regions of the nebula[6,7]. Precisely the same self-shielding effect occurs for C isotopes, with saturation of $^{12}C^{16}O$ and unsaturated $^{13}C^{16}O$ lines, resulting in C atoms highly enriched in $^{13}C$. However, it is less clear that $^{13}C$ enrichment can be retained and sequestered in nebular materials. C atoms are ionized in a continuum at wavelengths ≤110 nm, so that ionization of atomic C accompanies CO dissociation (unless C is optically thick). Once ionized, a rapid ion-molecule exchange reaction[32], $^{12}CO + ^{13}C^+ \leftrightarrows ^{13}CO + ^{12}C^+ + 35$ K, acts to erase any $^{13}C$ excess in C$^+$ at high temperatures, and at lower temperatures produces enrichments in $^{13}CO$ and $^{12}C^+$. CO photolysis also yields C atoms in the C($^1$D) excited electronic state. About 90% of CO photodissociation proceeds by the spin-allowed reaction, $CO + h\nu \rightarrow C(^3P) + O(^3P)$. Velocity-map imaging measurements show that approximately 8% of CO follows a spin-forbidden pathway[33], $CO + h\nu \rightarrow C(^1D) + O(^3P)$ (see "Methods"). The C($^1$D) is highly reactive, forming CH radicals upon collision with H$_2$, and therefore needs to be included in chemical models of the nebula.

**Chemical timescales of C loss in the solar nebula.** In order to assess possible chemical pathways for C species in the solar nebula, we estimate chemical loss timescales for the most relevant species due to a variety of reactions. We compute loss timescales at the FUV surface where optical depth is approximately unity for $^{12}C^{16}O$. We employ a simple, analytical model for the number density of H nuclei in the solar nebula[34]. The FUV optical depth of CO is given by

$$\tau_{CO}(\lambda, R, Z) = \sigma_{CO}(\lambda) f_{CO} N_H(R, Z), \quad (2)$$

where $\sigma_{CO}$ is the CO absorption cross section, and $f_{CO}$ is the fraction of CO gas in the nebula. We assume the CO fraction is constant with $Z$ with the value $f_{CO} = 2 \times 10^{-4}$; this assumption

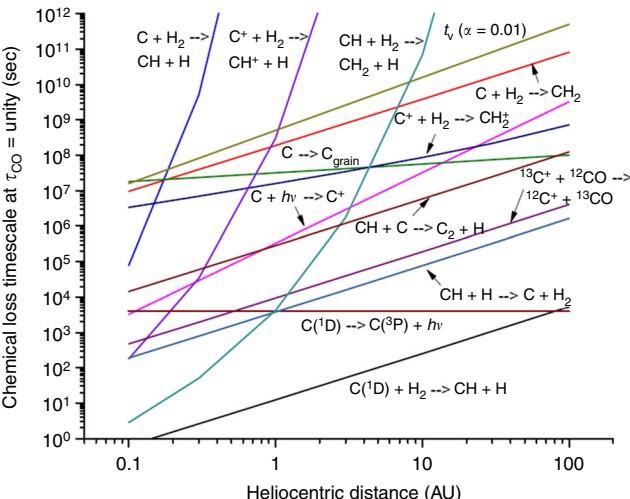

**Fig. 7** Chemical loss timescales for several carbon species computed at the FUV surface of the nebula and as a function of heliocentric distance. The figure shows that excited state carbon, C($^1$D), is lost to reaction with H$_2$ faster than for any other loss pathway. The C ionization reaction to form C$^+$ is FUV radiation from the protosun; nearby O stars can also be a significant source of FUV radiation. The timescale for the reaction CH + H → C + H$_2$ was computed assuming an H/H$_2$ fraction = 0.01. This reaction returns C atoms to the gas in their ground state, allowing ionization and exchange to remove the C isotope self-shielding signature. Depending on the degree of dust settling and the CO fraction, the H/H$_2$ ratio can be higher or lower by a factor of ~10$^2$–10$^3$ (ref. [66]). A much lower H/H$_2$ ratio allows the reaction CH + C → C$_2$ + H to sequester the self-shielding signature in C$_2$. Reaction of CH with H$_2$ also becomes much faster inside of 1 AU, providing another possible pathway for sequestering a C isotope self-shielding signature in larger molecules

allows an analytical evaluation of the vertical column density ("Methods"). The resulting values for $Z$ at unit optical depth (in the vertical direction) are shown in Fig. 6 for a vertically isothermal nebula.

For a 2nd-order reaction, A + B → C + D, the chemical loss timescale for species A is given by

$$t_A = \frac{1}{k_{A+B}[B]}, \qquad (3)$$

where $k_{A+B}$ is the two-body rate coefficient, and [B] is the concentration of B. For 1-body reactions, such as photodissociation or radiative relation, described by A → B, the loss timescale is $t_A = 1/k_A$. Figure 7 shows the loss timescales for several carbon species, including C, C$^+$, and several reactions for CH. One key point of the figure is that excited state carbon, C($^1$D), reacts rapidly with H$_2$ to form CH. This allows C($^1$D) from CO photolysis to avoid photoionization and exchange of C$^+$ with CO, which erases the isotope self-shielding isotope signature in the liberated carbon atoms. The chemistry of molecular carbon is quite complex in the solar nebula, and the fate of CH must be evaluated in a complete nebular chemical model. It is also not clear that a 5–10% production of C($^1$D) can play a significant role in altering enriching the $^{13}$C composition of planetary material; again, detailed models are needed to evaluate this.

Also shown in Fig. 7 is the vertical mixing timescale,

$$t_v \sim \frac{H_v^2}{D_t}, \qquad (4)$$

where $H_v$ is the vertical scale height of the solar nebula, and $D_t$ is the turbulent diffusivity in the nebula. It has been standard to

assume a turbulent diffusivity driven by magnetorotational instability (MRI), for which $D_t \sim \alpha c_s H_v$, where $c_s$ is the sound speed and $\alpha$ parameterizes the strength of turbulent mixing and has a value generally ~10$^{-4}$ to 10$^{-2}$. Figure 7 shows $t_v$ for $\alpha = 10^{-2}$. Recent work[35] argues that the MRI is actually too dissipative to yield such vigorous turbulent mixing throughout the disk, so the plotted $t_v$ represents a maximum plausible degree of turbulence in the bulk nebula (see "Methods").

The chemical reaction timescales in Fig. 7 show that reaction of C($^1$D) with H$_2$ is faster by a factor ~10$^3$ than the next fastest process, radiative relaxation, in the inner solar nebula, and is comparable in timescale to radiative relaxation beyond ~50 AU. Subsequent reactions of CH may lead to sequestration of C in larger molecules, e.g., by the reaction CH + H$_2$ → CH$_2$ + H, or by CH + C → C$_2$ + H, with possible preservation of the CO self-shielding isotope signature in the products. However, CH also reacts rapidly with H atoms, which returns C to its ground state, leaving it susceptible to ionization and exchange. The fate of CH is thus strongly dependent on the local H/H$_2$ ratio. A full assessment of the effects of CO self-shielding on C isotopes requires detailed nebular chemistry modeling, including atomic excited states, but the results of Fig. 7 suggest that preservation of $^{13}$C enrichment in C atoms liberated by CO photodissociation may not necessarily occur at the surface of the nebula. Carbon isotopologues of CO in young protoplanetary disks often show a large depletion in $^{13}$CO in the gas phase, as expected for CO self-shielding, but the depletion is not generally correlated with depletions in the $^{12}$C$^{17}$O and $^{12}$C$^{18}$O isotopologues[31]. The lack of correlation is possibly a result of C$^+$ formation, followed by exchange with CO, as discussed above. Thus, a self-shielding signature is very likely present in C isotopes initially, but is erased by exchange of C$^+$ and CO.

It has been suggested that the $^{13}$C depletion observed in some protoplanetary disks may be a result of CO ice formation[31]. We briefly consider whether CO ice formation in the outer nebula can explain the difference between solar and planetary $\delta^{13}$C. Formation of a large fraction of CO ice, enough to account for a ~50‰ shift in $\delta^{13}$C in CO gas in the inner solar nebula, would also produce a mass-dependent fractionation in O isotopes. Assuming the CO ice remains sequestered in the outer solar nebula, the residual gas-phase CO in the nebula would have been mass-dependently shifted. In order to be consistent with meteoritic and solar photosphere data, the CO would have to be mass-dependently shifted onto the CAI mixing line (Fig. 8). This is required to explain the O isotope data of secondary magnetites[36], which provide direct meteoritic evidence for CO self-shielding and formation of H$_2$O from the product O atoms. We assume that isotope fractionation due to CO ice formation implies bulk CO (gas + ice) isotope ratios in the parent molecular cloud that are mass-dependently shifted by ~ +50‰ in $\delta^{13}$C and $\delta^{18}$O relative to solar, if this process is to explain meteoritic data. Observations of $^{12}$CO and $^{13}$CO in protoplanetary disks and cloud cores do not clearly demonstrate isotopically enriched CO ice and depleted gas[31]. Instead, both CO gas and ice show $^{12}$C/$^{13}$C ~60–160, with no clear anti-correlation in the two reservoirs, so the role of CO ice formation in protoplanetary disks is uncertain. About 50–100‰ shifts are easily incorporated within this large range of $^{12}$C/$^{13}$C. But more problematic is that pre-existing nebular water would have to also need to reside near the O isotope CAI mixing line, as a result of water ice formation or the initial reservoir ratios of O isotopes (Fig. 8). Such a coincidence seems unlikely, and we therefore discard CO ice formation as an explanation for the $\delta^{13}$C difference between the Sun and terrestrial planets.

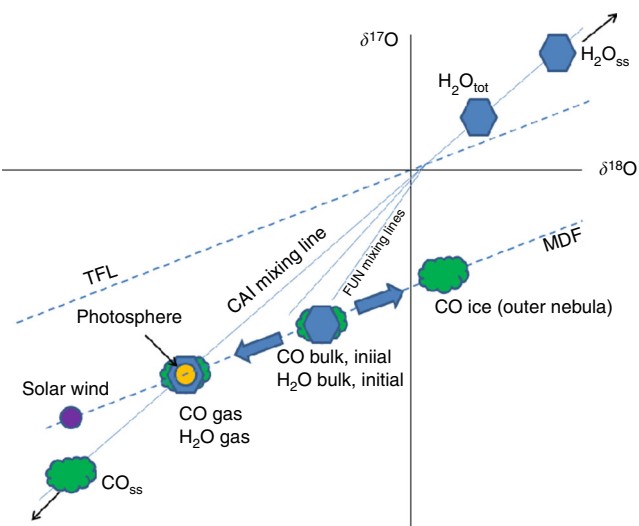

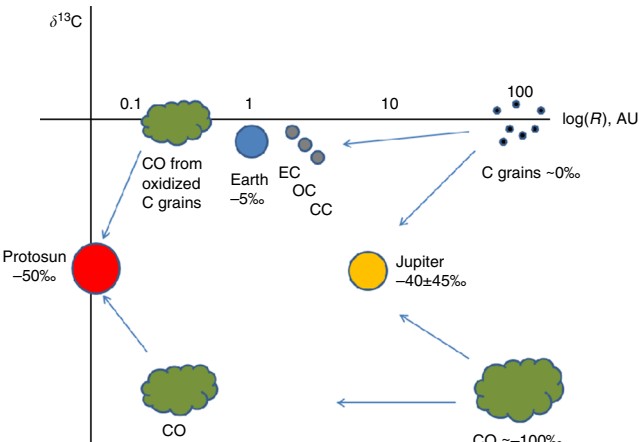

**Fig. 8** An illustration of how CO condensation in the outer solar nebula could qualitatively affect O isotope reservoirs. Assuming that condensed CO remains in the outer nebula, and is not mixed back into the O reservoir from which planets were formed, then there will be a mass-dependent enrichment in the CO ice relative to the bulk initial CO from the parent cloud. If CO self-shielding yielded $H_2O$ at the other end of the mixing line ($H_2O_{ss}$), the dissociated CO gas must have been on or close to the CAI mixing line. Residual (undissociated) CO gas ($CO_{ss}$) would have been highly depleted in $^{17}O$ and $^{18}O$ (refs. [7,31]). If the initial bulk reservoir of $H_2O$ was isotopically similar to CO initial bulk, then it too must have been mass-dependently fractionated until it resided on or near the CAI mixing line. This seems an unlikely occurrence, suggesting that perhaps the CO condensed phase was not large enough to substantially alter the O (and C isotopes) of the CO and from which planets were formed. Mixing lines for FUN inclusions are also shown, but the mass-dependent displacement that defines the lower end of the FUN inclusion mixing lines[67] are mostly likely derived from melting and evaporation of precursor CAIs[68], and not from the bulk initial CO and $H_2O$ reservoirs

**Fig. 9** Illustration of one possible explanation for the difference between solar and terrestrial planet and asteroidal C isotopes. If, in the parent molecular cloud, carbonaceous grains have a $\delta^{13}C_{PDB}$ values ~0‰ and CO has $\delta^{13}C_{PDB}$ ~−100‰, and C is approximately equally distributed between grains and CO gas, then formation of Jupiter from both grains and gas would yield $\delta^{13}C$ ~−50‰. Terrestrial planets and asteroids, formed primarily from planetesimals with C from C grains, would have $\delta^{13}C$ ~0‰. If C grains were converted to CO gas in the hot, inner solar nebula[69], the total accreted CO gas would yield a solar $\delta^{13}C$ ~−50‰, as reported here. The cause of the C isotopic ratio difference in grains and dust of the parent cloud may have been a result of self-shielding in the parent cloud exposed to a high FUV radiation field, as, e.g., observed in Ophiucus[37]

### Inheritance of terrestrial planet C isotopes from the parent cloud.

The chemical loss timescales illustrated in Fig. 7 suggest that enrichment of $^{13}C$ in planetary materials due to CO self-shielding in the solar nebula may not have occurred. Yet $\delta^{13}C$ depletion is often observed in molecular clouds, and has been attributed to self-shielding by $^{12}C^{16}O$ (e.g., ref. 37). We suggest here a simple scenario for inheritance of C grains enriched in $^{13}C$ from the parent molecular cloud from which the solar system formed. Grains that accreted to the outer nebula during infall would likely not be isotopically altered, or may have even been slightly enriched, during heating in the accretion shock. Accumulation of these grains in planetesimals just beyond the snow-line, and subsequent delivery of $H_2O$ and C by these planetesimals, could have provided Earth with its observed C isotope reservoir[38]. This scenario must also account for the depletion in C in the inner solar system relative to solar elemental values. The high C depletion in terrestrial planets (~$2 \times 10^{-4}$ for Earth relative to solar), and the approximate positive gradient with distance of C concentration in enstatite, ordinary and carbonaceous chondrite meteorites, has been attributed to chemical erosion of carbon grains in the inner solar nebula by reactions with H, OH and/or O[39,40]. Mass-dependent fractionation of C isotopes will likely accompany these reactions, but at the high temperatures needed for graphite erosion (~1000 K), isotope fractionation will be small, and we therefore neglect C isotope fractionation associated with C grain oxidation in the inner solar nebula. Oxidation of C grains in the hot inner solar nebula would produce CO with a $^{13}C/^{12}C$ ratio similar to the grains (assuming complete conversion of grains to gas), suggesting that initial nebula CO was isotopically lighter than the Sun (Fig. 9).

### Discussion

We have determined the C isotope ratio of the solar photosphere by reanalyzing shuttle-borne ATMOS FTS data collected in the mid-1990s. Using a new DMF for ground state CO, we obtain a photospheric $^{18}O/^{16}O$ ratio consistent with the inferred value from Genesis solar wind measurements[1,41]. The corresponding $^{13}C/^{12}C$ ratio is light, with $\delta^{13}C = -48 \pm 7$‰ VPDB. Using the ICD model we predict a solar wind $\delta^{13}C$ ~−70 to −90‰ for $C^{6+}$ and $C^{5+}$ coronal ions, respectively. Our solar wind predictions for coronal $C^{5+}$ agree with the range reported from ion microprobe analyses of solar wind implanted in lunar grain silicates[8], but disagree with $\delta^{13}C = 1 \pm 7$‰ from TiC grains in CAIs from the Isheyevo meteorite[9]. An examination of chemical loss timescales (in lieu of a full chemical model) of C isotope fraction due to CO photodissociation and self-shielding in the surface solar nebula, suggests that preservation of a $^{13}C$ enrichment in planetary materials may not have occurred. We offer an alternative interpretation of inheritance of the $^{13}C$-enriched C grains from the parent molecular cloud, again derived from self-shielding of CO. We have not explored here the composition of the carbon-carrying grains, but both irradiated ices and polycyclic aromatic hydrocarbons are plausible carriers.

The Coulomb drag theory neglects the potential role of magnetohydrodynamic (MHD) waves in the solar atmosphere in modifying isotope fractionation during solar wind formation. Given the likely importance of MHD waves in heating the upper chromosphere and corona, it is possible that other isotope fractionation processes could be present[42]. Evidence for chromospheric heating by torsional Alfvenic waves has been found from Solar Dynamics Observatory observations by McIntosh et al.[43]. Elemental fractionation is well established as a function of first

ionization potential (the FIP effect), and has been attributed to a ponderomotive force associated with Alfven wave propagation[44]. This force appears incapable of mass-independently fractionating isotopes of the same ion, but mass-dependent effects may be important.

Finally, we note that if self-shielding by CO in the solar nebula and/or in the parent cloud is responsible for the formation of $^{17}O$ and $^{18}O$-rich water and $^{13}C$-rich C grains in quantities needed to enrich terrestrial planet isotope ratios relative to the Sun, then the chemical mass-independent fraction schemes[45] involving SiOH or SiO are not necessary. Mass-independent fractionation of O isotopes in SiOH reactions presumably involve non-statistical effects, which are thought to be responsible for the mass-independent isotope effects seen in $O_3$ formation[46]. Similar non-statistical effects in the chemical reactions that form C grains are unlikely to produce enrichment in $^{13}C$ because of the complex set of pathways and products leading to grain formation.

## Methods

**Construction of co-added CO absorption lines.** CO lines were co-added (hybridized) as described in Ayres et al.[12]. The full-width half-max of the CO absorption lines is ~4.4 km s$^{-1}$, approximately the thermal Doppler broadening. All lines with a $^{12}C^{16}O$ line center within ~10 km s$^{-1}$ were rejected. Saturated lines were also rejected. Co-added lines were constructed by combining lines of similar frequency and $E_{low}$. This was done to improve individual line SNR, but also to lessen the total computation time needed for radiative transfer calculations. For $^{12}C^{16}O$, 36 hybrid lines (10 for $\Delta v = 1$, and 26 for $\Delta v = 2$) were constructed from 150 input transitions. For $^{13}C^{16}O$, $^{12}C^{17}O$, and $^{12}C^{18}O$, 9, 4, and 4 hybrid lines, respectively, were constructed from 70 observed transitions. The O abundance was computed for a given solar model using the $^{12}C^{16}O$ hybrid lines by adjusting the O abundance to reproduce the observed equivalent widths. Rare isotope abundances were computed for a given O abundance and solar model, by adjusting the isotopic abundances to match the observed isotopic equivalent widths.

**Solar model and radiative transfer.** The solar atmosphere and radiative transfer models closely follow Ayres et al.[12]. The CO5BOLD radiative hydrodynamic model was used to simulate the solar atmosphere[13,22], and uses 12 opacity bins and a Rosseland mean absorption. Full 3D simulations were required to reproduce continuum observations from $440 - 680$ nm, with the continuum optical depth unity in the visible and IR at ~0.1 bars, deeper than the peak in CO at ~$10^{-3}$ bars. Instantaneous equilibrium chemistry (ICE) has been previously shown to accurately simulate the composition of the middle photosphere[47], and is used here. The neutral composition is dominated by atomics, H, O, C, and N, with CO the most abundant molecular species in the middle photosphere (at the CO peak near $10^{-3}$ bars, CO is 38% of the total carbon, with the rest atomic C). Radiative transfer modeling is used with isolated columns from the 3D model, but with angle-dependent visible scattering[11,12,48]. A Feautrier-based Hermitian solution scheme is employed[49], with a black-body source term. With continuum source functions for each 3D column, specific intensities were computed along rays, reproducing the observed center-to-limb intensity variation. CO number density was calculated using ICE, and the O mole fraction was computed assuming $\varepsilon = 2\varepsilon_C$ (ref. [23]). CO line opacities were computed assuming thermal Doppler broadening, and for LTE conditions which are valid for CO in the middle photosphere[50]. $^{12}C^{17}O$ features are weak and are susceptible to errors in the local continuum level.

**Calculation of new oscillator strengths.** Oscillator strengths are computed with equation (1). The dipole moment matrix elements are given as

$$M_{v''J''}^{v'J'} = \langle v'J'|\mu(r)|v''J''\rangle \qquad (5)$$

where $\mu(r)$ is the electric dipole moment at internuclear distance $r$. Expressing the dipole moment in a polynomial expansion of the deviation from the equilibrium internuclear distance, $x = (r - r_e)/r_e$,

$$\mu(r) = \sum_{i=0}^{n} M_i x^i \qquad (6)$$

and the matrix elements become[16]

$$M_{v''J''}^{v'J'} = \sum_{i=0}^{n} M_i \langle v'J'|x^i|v''J''\rangle \qquad (7)$$

Li et al.[16] used Level 8.2 software[51] to determine the expectation values $\langle v'J''|x^i|v''J''\rangle$ and the coefficients $M_i$. In the present work we computed the dipole

moment matrix elements from the $M_i$ values from Li et al., and our line positions were computed from HITEMP[21].

**Inefficient Coulomb drag.** The O isotope values measured in the solar wind samples collected by Genesis, $\delta^{17}O = -80.8‰$ and $\delta^{18}O = -102.3‰$ (ref. [1]), differ from those of the photosphere due to isotopic fractionation during formation of the solar wind[3]. The most well studied process is ICD in which protons collide with highly ionized heavier elements in the solar corona, transferring momentum to the heavier ions after repeated collisions. ICD is most important in the 'slow' solar wind regime (<500 km s$^{-1}$), and derives primarily from the gravitational separation of ions by mass. In the 'fast' solar wind, ICD is not an important source of fractionation. Neglecting pressure gradients, thermal diffusion, and Alfven wave forcing, the one-dimensional ion momentum equation due to collisions, gravity, and the radial ambipolar electric field, can be written as[3]

$$m_x \nu_{xp}(u_x - u_p) = -\frac{(2A_x - Q_x - 1)}{2} \frac{GM_{sun} m_p}{r^2}, \qquad (8)$$

where $m_x$ and $m_p$ are the masses of the heavy ion and proton, respectively, $\nu_{xp}$ is the ion–proton collision frequency, $u_x$ and $u_p$ are the vertical velocity components for ions and protons, $A_x$ and $Q_x$ are the atomic mass and charge of the ions, and $r$ is the heliocentric distance of the ion in the corona. The ion-collision frequency is proportional to $Q_x^2/m_x$ (full expression is given in ref. [42]). Solving for the ion velocity[3],

$$u_x = u_p \left(1 - H_x \frac{C_p}{p}\right), \qquad (9)$$

where $C_p$ is a constant dependent only on the proton temperature, $\Phi_p = n_p v_p r^2$ is the proton flux integrated over full space, and the Coulomb drag factor $H_x$ is

$$H_x = \frac{2A_x - Q_x - 1}{Q_x^2} \sqrt{\frac{A_x + 1}{A_x}} \qquad (10)$$

Isotopic fractionation of heavy ions in the fast and slow solar wind may be computed from the respective ion velocities for these two solar wind regimes. For two heavy ions, $i$ and $j$, the fractionation factor $f_{i,j}$ is the ratio of the velocities

$$f_{i,j} = \left(1 - H_i \frac{C_p}{p}\right) / \left(1 - H_j \frac{C_p}{p}\right) \qquad (11)$$

Because the factor $C_p/\Phi_p$ is not accurately known, measurements of a known fractionation, in this case H and $^4$He, are used to calibrate Coulomb drag fractionation between other ions. The equation for the fractionation factor may then be expressed as[52]

$$f_{i,j} = \frac{H_{^4He} - H_i(1 - f_{He,H})}{H_{^4He} - H_j(1 - f_{He,H})}, \qquad (12)$$

where $f_{^4He,H}$ is the ratio of $^4$He/H (mole fraction) in the slow solar wind to that in the fast solar wind. The validity of the ICD model has been established through measurements of noble gas isotopes (He, Ne, Ar) from Genesis samples of slow and fast regime solar wind. The photosphere value is taken to be the bulk solar value of 0.084, as determined by helioseismology[53]. The measured bulk solar wind value of $^4$He/H = 0.0402 with coronal mass ejection (CME) material included, and $^4$He/H = 0.037 if CMEs are not included[52]. This implies $f_{^4He,H} = 0.4786$ with CMEs, and $f_{^4He,H} = 0.4405$ not including CMEs. Because the Genesis O isotope data was collected with a bulk solar collector, we use the values that include CMEs. In the equation for the fractionation factor, $H_{^4He} = 1.3975$.

The resulting ICD parameters for several charge states of O and C in the solar corona are given in Supplementary Table 1 for 3 charge states of O and C. The mean charge state for O is thought to be $Q = +6$ in the acceleration region of the corona[42]. For C the mean charge state is thought to be $Q = +5$. The isotopic fractionation in permil is computed as $100(f_{i,j} - 1)$. For $Q = +6$, $^{18}O$ and $^{17}O$ in the photosphere are enriched by 60 and 29‰ relative to $^{16}O$ in the bulk solar wind measured by Genesis (Fig. 4). McKeegan et al.[1] computed similar values from the ICD model, but then scaled their results to intersect the CCAM line; we are presenting the ICD model results without scaling. For $Q = +5$, $^{13}C$ is predicted to be depleted by 44‰ relative to $^{12}C$ compared to the photosphere value determined here (Fig. 5).

It should be recognized that many uncertainties are present in both the isotopic and elemental fractionation in the acceleration region of the corona[52]. In particular, it is unclear how well correlated isotope fractionation in the corona is to the elemental fractionation of $^4$He relative to H. Also, the charge state of a given ion in the corona is uncertain. For these reasons, the ICD results presented here are meant as an approximate guideline to the magnitude of isotope fractionation in the corona.

**FUV surface of solar nebula**. We employ an analytical model for the number density of H nuclei in the solar nebula[34],

$$n_H(R,Z) = n_0 \left(\frac{R}{100}\right)^{-2.75} e^{-a\frac{Z^2}{2R^3}} \qquad (13)$$

$$T(R,Z) = T_0 \left(\frac{R}{100}\right)^{-0.5} \left(\frac{L}{L_{sun}}\right)^{0.25} \qquad (14)$$

where $n_0 = 1.9 \times 10^9$ H nuclei cm$^{-3}$, $R$ and $Z$ are the heliocentric distance and height above the midplane in AU, and $a = \frac{GM_{sun}\mu m_H}{kT(R)}$. In the expression for $a$, $\mu = 2.37$ is the mean molecular mass of the nebular gas, and $m_H$ is the mass of the H atom. $T$ is the vertically isothermal gas temperature with $T_0 = 28$ K, and $L$ is the protosolar luminosity, which we assume here to be equal to $L_{sun}$. Equation 13 is an approximate form of the expression given by Aikawa and Herbst[34], and is valid for $Z^2 \ll R^2$. At $R = 100$ AU, $T = 28$ K and $a = 9055$ AU, a value we define as $a_0$. The exponential in equation 13 may be written as $e^{-0.5b(R)Z^2}$, where

$$b(R) = a_0 \left(\frac{R}{100}\right)^{0.5} R^{-3} = 905.5 R^{-2.5} \qquad (15)$$

where $R$ has units of AU and $b$ has units of (AU)$^{-2}$. To avoid a two-dimensional radiative transfer problem, we assume FUV radiation arriving normal to the disk midplane, even if it derives from the protostar. (This is a commonly made assumption[7,32]). The column density of H nuclei from the top of the nebula down to a height $Z$ is then

$$N_H(R,Z) = \int_Z^\infty n_H(R,Z')dZ' = n_0 \left(\frac{R}{100}\right)^{-2.75} \sqrt{\frac{\pi}{2b(R)}} \mathrm{erfc}\left(Z\sqrt{\frac{b(R)}{2}}\right) \qquad (16)$$

where "erfc" is the complementary error function with its usual definition, $\mathrm{erfc}(x) = \frac{2}{\sqrt{\pi}}\int_x^\infty e^{-t^2}dt$. The FUV optical depth of CO is given by

$$\tau_{CO}(\lambda,R,Z) = \sigma_{CO}(\lambda)f_{CO}N_H(R,Z) \qquad (17)$$

where $\sigma_{CO}$ is the CO absorption cross section, and $f_{CO}$ is the fraction of CO gas in the nebula. Neglecting photodissociation of CO, we assume the CO fraction is constant with $Z$ with the value $f_{CO} = 2 \times 10^{-4}$. Solving this set of equations for $Z$ at CO optical depth unity at a given wavelength, we find

$$Z = \sqrt{\frac{2}{b(R)}} \mathrm{erfc}^{-1}\left(\frac{\left(\frac{R}{100}\right)^{2.75}\sqrt{\frac{2b(R)}{\pi}}}{\sigma_{CO}(\lambda)f_{CO}n_0}\right) \qquad (18)$$

The resulting values for $Z$ are shown in Fig. 6. A similar analytical formulation was used by the first author in Antonelli et al.[54].

**Dissociation probabilities for C($^1$D) formation**. The spin-forbidden reaction CO $+ h\nu \to$ C($^1$D) $+$ O($^3$P) arises due to coupling of a $^1\Pi$ excited state to a $^3\Pi$ valence state at wavelengths less than 100 nm[33]. The fraction of CO that yields C($^1$D) during photodissociation, $\chi_{C(1D)}$, may be approximated as

$$\chi_{C(1D)} = \sum_{i=1}^{33} \phi_i^{C(1D)} \phi_i f_i \bigg/ \sum_{i=1}^{33} \phi_i f_i \qquad (19)$$

where $\phi_i^{C(1D)}$ is the branching ratio for C($^1$D) production[33], and $\phi_i$ is the quantum yield for dissociation[55,56]. Using the CO band numbers defined in van Dishoeck and Black[55], only bands 1, 3, 8, 16, 17, 20, 22–25 contribute to C($^1$D) production, yielding an 8.5% fraction of C($^1$D) from 91.2 to 108 nm. Absorption by H and H$_2$ has been neglected in this estimate, and a uniform (interstellar-like) FUV radiation field has been assumed.

**MRI and disk winds**. Previous models of self-shielding of CO in the solar nebula have assumed vigorous turbulent mixing associated with the MRI. The potential importance of non-ideal MHD in protoplanetary disks has been recognized in the past few years[35]. Non-ideal MHD limits vigorous turbulent mixing due to the MRI to the FUV-active surface layer of the disk. In this active layer $\alpha \sim 10^{-1}$ to $10^{-2}$ is plausible. Deeper in the disk, a lower turbulent viscosity parameter is predicted, $\alpha \sim 10^{-4}$–$10^{-3}$. This has two implications: (1) CO self-shielding in the outer solar nebula may become too slow to explain the enrichment in solar system O isotopes[57]; instead, self-shielding in the parent cloud core or in the inner nebula would have to be invoked[6,40]; (2) outer solar system disk winds become a significant source of mass loss[35]. These winds originate near the FUV surface of the solar nebula, and will preferentially carry away self-shielded CO (and N$_2$), i.e., gas enriched in $^{12}$C$^{16}$O and $^{28}$N$_2$. Disk winds provide a natural mechanism for enrichment of the solar nebula in the rare isotopes of N and O, and possibly of C as well. Loss of CO by photoevaporation or in disk winds is a necessary process for

removing CO highly enriched in $^{16}$O due to self-shielding from the solar nebula, although isotope fractionation during formation of tenuous disk winds is not likely to be significant.

**Data availability**. All relevant data are available from the authors.

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

## Acknowledgements

J.R.L. acknowledges support from the NASA Origins of Solar Systems program, grant NNX14AD49G to ASU. T.R.A. acknowledges support from NSF AST-0908293. Publication of this article was funded in part by the University of Colorado Boulder Libraries Open Access Fund.

## Author contributions

J.R.L. and T.R.A. conceived the project. E.G.N. and J.R.L. performed the oscillator strength calculations. T.R.A. performed the radiative transfer calculations. All authors contributed to writing the paper.

## Additional information

**Competing interests:** The authors declare no competing financial interests.

