## [Peer Review File · Nature Communications]

Reviewers' comments:

Reviewer #1 (Remarks to the Author):

"A light carbon isotope composition for the Sun"

Lyons and colleagues report new analyses of rovibrational spectra of solar CO incorporating new line strengths that provide the highest precision and, it is argued, the highest accuracy direct determinations of the photospheric oxygen-16/oxygen-18 ratio. The results quantitatively confirm prior inferences of solar oxygen based on measurements of the solar wind by the Genesis mission. Additionally, Lyons et al. extend their analysis to make a prediction about the solar carbon isotopic ratio, which is presently not well-constrained. These represent important results of keen interest to the wide astronomical and planetary science communities, and thus the work is highly appropriate for publication in Nature Communications. A concern, however, is that the discussion of the calculation is highly technical and some key points are not well explained, even in the supporting materials. This reviewer, at least, was not able to discern how sensitively the new isotope results depend on assumptions and/or parameters in the model.

Several points could be made clearer:

1. The 43K temperature "enhancement in middle photosphere": first, what is meant by "middle photosphere" ? second, how is this justified by the measurements? Finally, what is the isotope effect if this is wrong?
2. The computed O abundance is 'slightly low' compared to the value from helioseismology. What is the effect on isotope abundances? Is this reasonable?
3. Why are there differences in dipole moment functions (line 66 and following)? Or, said another way, what are the differences based on? How do these differences - of opposite sign for the fundamental and first overtone - translate into 6% difference in isotope ratio of $^{16}\text{O}/^{18}\text{O}$?

Related to the last point is the efficacy of Figures 1 and 2 in the main manuscript. It is not really explained what these are meant to show, nor is it clear how the interpretation of the trends of the data shown on either figure relate to the derived isotopic results (besides a statement that the change in dipole moment functions accounts for most of the isotopic shift). The figure captions and the text should explain better what the significance is of what is being plotted.

In my opinion, these shortcomings can be readily addressed by some rewriting or some modest additions in the text or the supplementary material.

The paper is very well written, but I did manage to find a few typos and places where some clarification of the prose should be considered.

line 16: consider putting a negative sign in front of 60

line 38: change meteorite to meteorites

line 138: change would to could

line 230,232 and figure legend(s): "v" is used but I think it should be ν ("nu")

lines 238, 240: same comment

figure 3: several terms on the figure are not explained (CAIs, CC, OC, terrestrial fractionation line). Note that hibonite is misspelled.

figure 4: there are several puzzling aspects of this figure and some improvements can be made. First, how is it known what heliocentric distance is appropriate for: EC, OC, CC, IDPs (inside of Jupiter?), 81P?. Second, how is the carbon isotope composition of Earth's core known? Third, it would be good to indicate the terrestrial mantle.

line 263: typo: change "form" to "from"

line 433: are there words missing after "mean molecular" ?

line 533: "Modeling C loss from grains..... chemical reactions." This is not a complete sentence.

line 641: "formation of" is repeated

finally, consider changing the title to "A light carbon isotope composition of the Sun" rather than "...for the Sun".

Reviewer #2 (Remarks to the Author):

I read with interest the new work by Lyons et al aimed at determining the carbon isotope composition of the solar photosphere. The stable isotope composition of the Sun is an important piece of cosmochemical information because the Sun represents the best available analogue of the composition of the protosolar nebula from which the planets formed. The stable isotopes of the light elements, H, C, N, O, present significant, sometimes extreme, variations among solar system objects and reservoirs. Understanding the causes of these variations may permit insight into cosmochemical components and processes that contributed and shaped the protosolar nebula.

Because the solar matter cannot be sampled directly, our knowledge of the elemental and isotopic composition of the Sun relies mainly on two different approaches. On the one hand, the spectroscopic analysis of the photosphere permitted to determine the elemental composition of many solar elements, but barely of the isotope compositions. On the other hand, the analysis of the solar wind (SW) gives insight into the isotopic composition of the major solar elements. The latter was the main scientific target of the Genesis mission, which collected SW ions during 27 months. The analysis of collected material permitted the determination of SW oxygen, nitrogen, and noble gas isotope compositions.

Carbon, the major forming element of organics, does not show large isotope variations in the solar system as do H and N, for reasons that remain to be determined. Nevertheless, given the astrophysical importance of this element and the fact that its two isotopes, ^{12}C and ^{13}C , show significant variations in the solar system, the determination of the solar C isotope composition is most welcome. So far, the only measurement of SW carbon isotope composition is that of Hashizume et al (2004) who proposed based on the analysis of lunar soil grains by ion probe that SW C has a $\delta^{13}\text{C}$ of ~ -100 permil. In passing this measurement should be clearly cited in the introduction of the present paper given its uniqueness. Thus this estimate supports enrichments of solar system objects and reservoirs in the heavy isotope ^{13}C , in parallel to those of N, H and O which also require contribution/fractionation enriching most reservoirs in the heavy and rare isotopes ^2H , ^{15}N , ^{17}O and ^{18}O . However, the Hashizume et al's result was for SW and not for the Sun itself, thus requiring to correct for SW fractionation, a process that is almost not known.

Lyons et al propose here another approach, based on the analysis of the photospheric emission (data obtained in previous experiments and available in the literature) using new deconvolution techniques that permit to identify O and C isotopes. Being not familiar with solar spectroscopy and its latest developments, I cannot comment on the techniques used here, except that one of the authors has published well received papers on the field.

The authors report their determination of the photospheric $^{18}\text{O}/^{16}\text{O}$ ratio which value of -50 ± 11 permil is consistent with what we know from Genesis. This is a good step to give confidence in their approach. They then get a $\delta^{13}\text{C}$ value of -48 ± 7 permil for the photosphere. They suggest that this value is representative of the Sun and of the protosolar nebula, arguing that other fractionation processes such as gravitational settling are not significant. They use the so called Coulomb drag model (which is challenged, but there is not much model available either) to estimate a SW range of -75 to -95 permil, consistent with the range of -90 to -120 permil determined by Hashizume et al for SW implanted in lunar soils. Finally the authors attempt to discuss the cosmochemical implications of their determination.

Overall this is an interesting piece of work and I recommend publication in Nature Comm, provided that experts in the field validate the photospheric deconvolution, and pending on significant improvement of the ms.

The ms. is not well prepared and suffers from several problems. Nature Comm allows full papers but this ms. seems to be a sequel submitted to a letters journal having drastic constraints on the length. Part of the interesting information on the used techniques is relegated in a Supplementary, as usual for letter journals, but not necessary here. Large pieces of information are somewhat hidden or not clearly expressed. For instance, in the Discussion section processes other than self shielding and presently shown into the Supplementary should be discussed in the main text. The discussion of these possible processes is oriented towards a public already aware of the positions and controversies in the field. Concerning the possibility of isotope fractionation in ice, could the authors precise what kind of fractionation process they expect, and what would be the direction of fractionation? If this fractionation also affected O as proposed, it should be mass-dependent according to the authors and therefore could not account for the $\delta^{18}\text{O}$ data. It seems that Lyons et al now prefer an origin of fractionation in the parent cloud. They should explain why the effect would be so large for N compared to C. I have also a difficulty with the Rayleigh distillation model applied to grain erosion. Grains are solid particles. When there is erosion, I doubt that the entire carbon of the grains would be in equilibrium with the fraction of lost C. I may have missed the nature of the process the authors propose, and I strongly suggest that they reformulate their discussion, by first clearly stating the starting processes and hypotheses for each case.

- In Figure 3, there are two data points with low error bars, labelled O7+ and O6+, that are not presented in the text (or that I missed as the text is unclear). I suppose that they correspond to different degrees of ionisation, but none of these are discussed. The way the O isotope composition is obtained is also mysterious for a public that has not followed closely the controversies on these topics in the last few years.

- In Figure 4, there are 3 data points pertaining to this study, one without label and the two others labelled with ionisation degrees. How is computed the non labeled? what is its ionization degree?

- Errors (eg abstract) should be precised eg 1 or 2 sigmas, 95 CI etc.

Reviewer #3 (Remarks to the Author):

This paper describes improved retrievals of the isotope ratios of oxygen and carbon in the Solar atmosphere as traced by archival high-resolution spectroscopy from the ATMOS experiment. While the data have previously been analyzed, this work presents a new retrieval using improved oscillator strengths for the rovibrational transitions of CO. The improved oscillator strengths remove, arguably, the dominant source of systematic uncertainty in the retrieved isotope abundance ratios of carbon and oxygen, and allows the authors to determine their values to greater accuracy than what was previously possible. It is important to note that it has been known for a long time that the CO oscillator strengths were uncertain, as different, independent (theoretical and measured) disagreed significantly, so this work addresses a long-standing problem by a timely application of these new oscillator strengths.

The methods applied to derive isotope ratios are apparently adopted from existing literature, e.g., by Ayres et al. 2013. This is a strength of the study, as it is shown that the single change of updating oscillator strengths reconciles the Solar spectroscopy with independent estimates of the Solar isotope ratios. However, a criticism of the manuscript, is that the text remains unclear as to which degree the Ayres et al. method is adopted in detail, or whether there are any departures (see comments below).

The authors find that the Solar composition is isotopically light, and consistent with the solar isotope ratios inferred from Genesis measurements of the Solar wind. This follows expectations, but lends great confidence to any derived interpretations of the data.

The study also finds a depleted $^{12}\text{C}/^{13}\text{C}$ ratio in the Sun, which is a new result, potentially consistent with nebular selective photodissociation.

Consequently, I think this is an important development in our understanding of the formation of the Earth and other Solar System bodies. The decades-long uncertainty in the Solar carbon and oxygen ratios has put a limit on the confidence with which we have been able to constrain the fractionation processes taking place in the Solar nebula. This work makes great progress toward solving the question of the Solar composition and its relation to nebular fractionation. My main concern relates to how confident we can be that the reported error bars are correct, as the manuscript in its present form does not provide sufficient detail. I believe the authors likely did the necessary work on quantifying their errors, and so should be able to provide an error budget and respond to my questions below without too much trouble.

Detailed comments:

- The study hinges on the claim that the errors on the inferred isotope ratios have been significantly decreased. However, I think that the derived errors are not sufficiently justified in the Supporting Material, mainly due to a relative lack of quantitative information about how they were derived. Specifically, the modeling procedure is very complex and will compound errors hierarchically. That is, there are data errors which lead to errors on the derived equivalent widths. The EWs are then used to compare to model with its own systematic errors, etc. While there is no doubt that the improved oscillator strengths will help, the manuscript does not make it clear how errors (statistical and systematic) are propagated. An approach such as described would likely lend itself well to a Bayesian approach, but that does not seem to be the case here.

What prior distributions were assumed for the error sources, and how were they propagated to arrive at the final posterior errors? Is it possible to report the full error budget that must have gone into calculating the reported errors? Can we reasonably assume that the posterior distributions are normally distributed (as suggested by the singular errors) and that there are no hidden degeneracies?

- The result appears to rely heavily on the spectroscopic methods described in great detail in Ayres et al. 2013. However, it is not clear from the text to which degree this work was exactly replicated, or whether there were any departures. Could the authors comment on that, so that the reader knows whether or not Ayres et al. 2013 is an appropriate reference for the methods in this work - and specifically which parts of the analysis are relevant (the text indicates it is the "analysis of ATMOS solar spectra", but is it also the fit of the atmospheric models?)

- It seems that the systematic error from the atmospheric model fit is based on the spread of inferred isotope ratios implied by different model assumptions. How do we know that the model grid reasonably samples the possible parameter space? Is the reader supposed to refer the discussion in Ayres et al. 2013 to justify the choice of model parameters?

- The C/O ratio is assumed to be exactly 0.5, but the referenced value (from Allende Prieto et al. 2002) has an uncertainty of 14%. How does this affect the error on the isotope ratios?

Replies to referees: A light C isotope composition for the Sun, by Lyons et al.

Before getting into the detailed replies, I would like to thank the 3 referees for their helpful reviews. I originally submitted this to Nature, and so the format was not appropriate for Nature Communications. The paper has been expanded, as per the Nature Communications format, and the points raised by the referees have been addressed. Figures have been added (there are now 9) to the main text, and the number of references has been increased. The *Methods* section contains most of the mathematical details.

Replies to the 3 referees are below, given in the order of the points raised.

Reviewer #1

“Several points could be made clearer”

1. What is the “middle photosphere”? The profile of enhanced temperature is from ~ 10 to 10^4 dyne cm^{-2} , and is illustrated in Figure 3 of Ayres et al. 2013. The top of the photosphere occurs at ~ 0.87 mbar (= 870 dyne cm^{-2}). So the range over which the temperature perturbation has been applied is from the ‘middle photosphere’ to the lower chromospheres. The text has been modified to indicate this (see p. 4 of revised text). Justification of this temperature perturbation comes from the requirement that $^{12}\text{C}^{16}\text{O}$ abundances are the same for $\Delta v = 1$ and $\Delta v = 2$ transitions. The improvement in spectroscopic line strengths (i.e. f -values) makes this constraint possible. The magnitude of the isotope effect that accompanies this constraint is illustrated in Table 1, column ‘Error 5’. Assuming that the revised f -values are correct, we get instead the errors indicated in ‘1- σ final’ column of Table 1.

2. “The computed O abundance is slightly low” compared to helioseismology. The error, if there is an error, in the elemental O abundance will have minimal impact on the derived isotope ratios. We have not quantitatively assessed this effect here. Is this reasonable? Yes, because all isotopologue abundances scale similarly, the change in isotopologue ratios is very small.

3. The difference in the f -values between Hure and Roueff 1996 (HR96) and Goorvitch 1994 (G94) is due to the different dipole moment functions assumed in these two papers. This is described in the 1st paragraph of p 4 of the revised manuscript.

“Efficacy of Figures 1 and 2”

Explanatory text has been added to the captions of both figures. Figure 1 is the raw observation data from ATMOS, but with some co-adding of lines. Figure 2 is the new oscillator strengths computed from Li et al., and is the basis for the new results reported in Figures 4 and 5.

‘Typos and clarification’ – note that nearly all line numbers have changed, but I will use the old line numbers given by the referee for the point raised.

line 16 – changed to -60 permil in abstract.

line 38 – done

line 138 – change would to could – I'm not sure which occurrence of 'would' the referee is referring to.

line 230, 232 – use ν instead of v . All occurrences of ' ν ' or ' v ' are the letter ν , not nu , and are the vibrational quantum number.

line 238, 240 – same as above.

Figure 3 (now Figure 4) – Definitions have been added to the figure caption for CAIs, CC, OC and terrestrial fractionation line. Spelling of hibonites has been corrected.

Figure 4 (now Figure 5) – The heliocentric distances for ECs, OCs, CCs, IDPs, and 81P are meant to be illustrative only, and not precise determinations. A sentence to this effect has been added to the caption. The C isotope composition of Earth's core is an estimate from models by Horita and Pulyakov (2015) (ref. 67). This reference was inadvertently left out of the original submission, but has been added here in the caption to (the new) Figure 5. A reference to the estimated fraction of C in the core (up to 90% of total Earth C) has also been added. Terrestrial mantle values have been explicitly labeled in Figure 5. I thank the reviewer for these substantial improvements to Figure 5.

line 263 – typo – change form to from - done

line 433 – change mean molecular to mean molecular mass.

line 533 – “modeling C loss from grains...” – this sentence has been removed.

line 641 – ‘formation of’ repeated – fixed.

Title: We've decided to keep 'for' in the title, but we do appreciate the suggestion.

Reviewer #2

The manuscript has been rewritten to the larger format allowed by Nature Communications. Much of the discussion that was previously in the Supplementary section is now in the main text. A Methods sections now contains most of the mathematical details.

“Concerning the possibility of isotope fractionation in ice...” – A short, qualitative section on isotope fractionation due to CO ice formation has been added to the manuscript. The affect of CO ice formation on O isotopes is also discussed in a new figure (Figure 9). A more quantitative treatment of the expected mass-dependent isotope fractionation will be presented elsewhere.

“It seems that Lyons et al. now prefer an origin of fractionation in the parent cloud” – I have always considered the parent cloud to be a possible site for CO self-shielding (e.g., see Ref. 40, authors Lee, Bergin and Lyons). The timescale analysis presented here for C loss in the surface of the nebula suggests that the parent cloud may be a better location for the origin of a large C isotope fractionation. From a

chemical standpoint, self-shielding in the nebula and parent cloud are very similar. However, the different number densities and pathlengths in the two environments mean that the preservation of photo-generated isotope affects can differ.

“why the effect would be so large for N compared to C” – This is a very important issue, but not one that we wish to address in this paper. To do so would require a detailed discussion of N₂ photolysis, which is difficult to do in a precise yet succinct manner.

“Rayleigh distillation” – We agree with the reviewer that the application of Rayleigh distillation to C grain erosion is not warranted. This text has been removed.

Figure 3 (now Figure 4) – O⁷⁺ and O⁶⁺ are ions of O in the solar corona, as described in the figure caption. Coulomb drag isotope fractionation is dependent on the charge state of the heavy ion (i.e., O⁺ or C⁺). A citation to key paper on this topic has been added to the caption. A more detailed discussion of coronal ions is given Coulomb drag section of Methods.

Figure 4 (now Figure 5) – The unlabeled red point is CO in the solar photosphere, and is unionized. The red points labeled C⁵⁺ and C⁶⁺ are the charge states for C in the corona and solar wind. The figure caption has been modified to make these points more clear.

“Errors” – Errors are 1- σ , which is now explicitly stated in the abstract. A sentence has been added to the text (2nd paragraph) indicating that all errors are 1- σ unless stated otherwise.

Reviewer #3

“It has been know for a long time that the CO oscillator strengths were uncertain...” – Yes, I discovered this after giving a talk on this topic. E. Roueff was in the audience, and came up afterward to thank me for the work. The real credit goes to Li et al. (2015) for determining the new DMF.

The solar atmosphere and radiative transfer models closely follow Ayres et al. 2013. The principal departure in the modeling presented here is the new oscillator strengths. A statement to this effect has been added to the Methods, but this should be readily apparent from the citations to Ayres et al. (ref. 12). The interpretation of the new C isotope ratio for the solar system is new and has been brought into the main text.

“The study hinges on the claim...”. The error analysis follows Ayres et al. 2013, but with a decrease in the error associated with the f-values. Table 1 has been added to list the various factors contributing errors. All errors are added in quadrature. Systematic errors are propagated in this error analysis only for the solar atmosphere temperature profile (see Fig. 3). Large systematic errors due to ¹²C¹⁶O line tail overlap with minor isotopologue (¹³C¹⁶O, ¹²C¹⁷O, ¹²C¹⁸O) peaks were eliminated in Ayres et al.¹². A Bayesian error analysis may be performed in the future, but would impose a significant burden on the authors for the present work.

“What prior distributions...”. The reviewer poses this question in terms of prior and posterior errors. We are not performing a Bayesian error analysis. All errors are assumed to be normally distributed.

“The result appears to rely heavily on...” As stated above, Ayres et al.¹² is an appropriate reference for the many of the methods in this work. That includes the solar atmosphere and radiative transfer models, but does not include the new f-values. Nor does it include the analysis of how this ¹³C enrichment could arise from CO self-shielding or other processes.

“It seems that the systematic error from the atmosphere model fit...” The reviewer is welcome to refer to Ayres et al.¹² for a discussion of the parameter space sampled by the atmospheric model.

“The C/O ratio is assumed to be exactly 0.5...” Allende-Prieto et al. 2002 give a C/O value of $0.50 \pm .07$. We have not run models with different C/O ratios, but our expectation is that the isotopic ratios will be only weakly dependent on the elemental ratios.

REVIEWERS' COMMENTS:

Reviewer #1 (Remarks to the Author):

The revised paper is well-written and the authors have made clear explanations regarding sources of uncertainties and have also provided an interesting discussion regarding implications of their finding. This is important work and I recommend publication.

one trivial suggestion:

Line 452. "correlated" is repeated (delete the first instance).

Reviewer #2 (Remarks to the Author):

This is a very good paper that gives an important cosmochemical information. The ms has been well re-arranged since the first submission. I suggest to publish the paper as it is, or after slight re-arrangement. I think the discussion about the origin of C isotope fractionation should go to the Discussion section. The units are in the CGS system, but I prefer the SI instead, this depends on the policy of Nature.

Reviewer #3 (Remarks to the Author):

The authors have provided a satisfactory response to my first report. I think the manuscript has been significantly improved, and have just a couple of minor comments. It would be good to see these revised solar carbon ratios, based on new oscillator strength, in the literature.

Potential routes for sequestration of ^{13}C into grains sufficiently refractory to be efficient carriers of enriched carbon to planetesimals in the inner solar system are not clearly discussed. For instance, while the authors indeed present a quantitative argument for the formation of persistent CH from enriched atomic carbon, it would be useful to consider candidates for the ultimate carbon carrier from CH into planetesimals. In particular, whether the ultimate origin of the carbon fractionation is in the prestellar or nebular phase could require different carbon carriers. In protoplanetary disks, it is known that complex carbon chemistry must be quite active, as both acetylene and PAHs are, if not ubiquitous, then at least common, in inner disks (Geers et al. 2006, Pascucci et al. 2009). This is not the case for dense clouds, and carbon is likely processed in irradiated ices. Is it possible that PAH chemistry plays a key role in the story of carbon enrichment of inner solar system?

Page 9: "We suggest here a simple scenario..." - Something seems to have been garbled in this sentence.

My responses to reviewers follow. Again, we thank the referees for their helpful comments.

Reviewer 1:

1. line 452 – We have removed the first occurrence of ‘correlated’ in this sentence.

Reviewer 2:

1. The reviewer recommends that we move the topic of C isotope fractionation into the Discussion section of the manuscript. However, this is a large portion of the manuscript, with multiple subheadings, so we feel that these sections should remain as they are in the main text of the manuscript.

2. The reviewer suggests that we use SI units throughout the manuscript. For the spectroscopic portion of the manuscript (1st half) we have used ‘wavenumbers’ (cm^{-1}) as the primary frequency unit, which is by far the most common usage for solar studies. For the disk portion of the manuscript (2nd half) we have used cgs units, which are the primary units used by the protoplanetary disk and solar nebula community. We have therefore retained our choice of units, and not switched to SI units.

Reviewer 3:

1. The reviewer points out that the carrier of C isotope signatures could be PAHs for the solar nebula scenario, or could be irradiated ices for the parent molecular cloud case. On page 10 we have added the sentence, “We have not explored here the composition of the carbon-carrying grains, but both irradiated ices and polycyclic aromatic hydrocarbons (PAHs) are plausible carriers.”

2. A preposition was missing from the sentence on page 9. An ‘of’ has been added, which greatly clarifies the meaning of the sentence. We thank the reviewer for pointing this out.

With regards,

James Lyons

SESE

ASU

USA

1-310-880-1992